# Concept for a Unidirectional Release Mucoadhesive Buccal Tablet for Oral Delivery of Antidiabetic Peptide Drugs Such as Insulin, Glucagon-like Peptide 1 (GLP-1), and their Analogs

**DOI:** 10.3390/pharmaceutics15092265

**Published:** 2023-09-01

**Authors:** Anubhav Pratap-Singh, Yigong Guo, Alberto Baldelli, Anika Singh

**Affiliations:** 1Food, Nutrition, and Health Program, Faculty of Land & Food Systems, The University of British Columbia, 2205 East Mall, Vancouver, BC V6T 1Z4, Canada; 2Natural Health and Food Products Research Group, Centre for Applied Research & Innovation (CARI), British Columbia Institute of Technology, Burnaby, BC V5G 3H2, Canada

**Keywords:** insulin, glucagon-like peptide 1, buccal delivery, diabetes, encapsulation

## Abstract

Injectable peptides such as insulin, glucagon-like peptide 1 (GLP-1), and their agonists are being increasingly used for the treatment of diabetes. Currently, the most common route of administration is injection, which is linked to patient discomfort as well as being subjected to refrigerated storage and the requirement for efficient supply chain logistics. Buccal and sublingual routes are recognized as valid alternatives due to their high accessibility and easy administration. However, there can be several challenges, such as peptide selection, drug encapsulation, and delivery system design, which are linked to the enhancement of drug efficacy and efficiency. By using hydrophobic polymers that do not dissolve in saliva, and by using neutral or positively charged nanoparticles that show better adhesion to the negative charges generated by the sialic acid in the mucus, researchers have attempted to improve drug efficiency and efficacy in buccal delivery. Furthermore, unidirectional films and tablets seem to show the highest bioavailability as compared to sprays and other buccal delivery vehicles. This advantageous attribute can be attributed to their capability to mitigate the impact of saliva and inadvertent gastrointestinal enzymatic digestion, thereby minimizing drug loss. This is especially pertinent as these formulations ensure a more directed drug delivery trajectory, leading to heightened therapeutic outcomes. This communication describes the current state of the art with respect to the creation of nanoparticles containing peptides such as insulin, glucagon-like peptide 1 (GLP-1), and their agonists, and theorizes the production of mucoadhesive unidirectional release buccal tablets or films. Such an approach is more patient-friendly and can improve the lives of millions of diabetics around the world; in addition, these shelf-stable formulations ena a more environmentally friendly and sustainable supply chain network.

## 1. Introduction

Diabetes mellitus is a chronic metabolic syndrome characterized by hyperglycemia that has become a global epidemic [1,2,3]. It leads to complications, including frequent urination, limb amputation, kidney failure, and cardiovascular disease, and even seriously threatens patients’ lives [4,5]. Diabetes mellitus can be divided into type 1 diabetes mellitus (T1DM) and type 2 diabetes mellitus (T2DM). T1DM accounts for about 5% of diabetes cases and is caused by insulin deficiency due to the apoptosis or loss of insulin-secreting pancreatic β-cells destroyed by the autoimmune system [6,7]. Unfortunately, a thorough cure for T1DM is still unavailable [8]. T2DM accounts for about 95% of diabetes cases. It is a complex metabolic disorder characterized by insulin resistance in the liver and muscle tissues or excessive hepatic glucose production associated with inappropriately high levels of glucagon [9,10]. Chronic hyperglycemia further impairs pancreatic β-cells, leading to other diseases. T2DM prevalence is increasing rapidly worldwide, and its risk factors include genetic predisposition, obesity, physical inactivity, and an unhealthy diet [11]. Lifestyle modifications, including weight loss, physical activity, and a healthy diet, are effective in preventing and managing T2DM. However, medication and insulin therapy can still be necessary for optimal glycemic control [12,13]. Early detection and intervention are crucial for preventing and managing diabetes and its complications, while regular blood glucose monitoring is highly recommended for all diabetes patients. Moreover, diabetes patients should receive comprehensive therapy to improve their quality of life and reduce the risk of complications, which includes diabetic education, nutritional counseling, and regular medical examinations [14,15]. Overall, diabetes is a serious and growing public health issue that requires collective efforts from healthcare providers, policymakers, and the general public to prevent, manage, and eventually cure the disease [16].

Currently, there are several medications available for patients with diabetes. Insulin is, without any doubt, the most common one. In type 2 diabetes, the body quits replying to normal insulin levels, and over time, the pancreas does not produce enough insulin to sustain the body’s needs. Therefore, introducing insulin in patients affected by diabetes can regulate glucose levels [17]. There are several types of insulin on the market; they differ mostly on the release time [18]. Insulin is not the only drug used for treating diabetes. Several alternatives are being raised, particularly in the literature references [19]. For instance, Rosenstock et al. showed a prosperous replacement of prandial insulin with the long-acting glucagon-like peptide 1 receptor agonist (GLP-1 RA) albiglutide in most (72%) tested patients with type 2 diabetes [20]. Further studies confirmed that substituting insulin with a few GLP-1 receptors improves glycemic control and reduces body weight and the risk for hypoglycemia [21]. Some of the main receptors available are exenatide, liraglutide, semaglutide, albiglutide, lixisenatide, dulaglutide, and beinaglutide. These GLP-1 receptors have been clinically tested; so far, only albiglutide has been approved by the Food and Drug Administration (FDA). However, the FDA recently granted Novo Nordisk the oral delivery of semaglutide [22].

Although peptide drugs have gained attention in diabetes mellitus therapy due to their potential efficacy, their clinical use is limited by the challenges of subcutaneous injection [23]. Most commercially available insulin or GLP-1 receptors are delivered through an intracutaneous injection. Due to the patient’s discomfort, alternative delivery routes have been exploited. The oral route is possibly the most common and the most acceptable for patients [17,24]. The oral administration of insulin or GLP-1RA can mimic the dynamics of the endogenous hormones and provide better glucose homeostasis. However, oral delivery faces challenges such as low permeation, proteolysis enzymes in the upper gastrointestinal tract, low gastric pH, and bioavailability [17]. Furthermore, the gastrointestinal route is unpredictable and usually involves high losses of the delivered drug [17,25]. Therefore, the highest number of current references are focused on defining an innovative route of delivery. The inhalable version of insulin was developed a few decades ago, but a new form has recently been approved by the FDA [26]. The main issues encountered with this drug are acute bronchospasm in patients with asthma and chronic obstructive pulmonary disease (COPD), hypoglycemia, cough and throat pain/irritation, and a significant decrease in the diffusion capacity of lungs [27]. Overcoming these challenges is crucial to achieving the potential benefits of oral peptide drugs in diabetes mellitus treatment.

Recent research studies have focused on developing new drug delivery systems, such as nanoparticles and liposomes, to protect peptide drugs from degradation in the gastrointestinal tract and enhance their absorption [17,28]. Additionally, modifications to the peptide structure, such as the use of prodrugs, have shown promise in improving the stability and bioavailability of oral peptide drugs. Moreover, the development of new formulation technologies, such as spray drying and freeze drying, has enabled the production of stable and easily administered oral peptide drug formulations [18,29]. Furthermore, advances in nanotechnology and biotechnology have allowed for the design of novel drug delivery systems, such as implantable devices and micro-needles, that could provide a sustained release of peptide drugs and eliminate the need for frequent dosing [30,31]. These innovations offer hope for the development of effective and convenient oral peptide drugs for diabetes mellitus therapy. However, their clinical translation requires comprehensive safety, efficacy, and feasibility testing. Regulatory approval for these new drug delivery systems will also require their superiority over existing therapies to be demonstrated and any potential side effects to be addressed. Despite these challenges, the potential benefits of oral peptide drugs in diabetes mellitus treatment make it a promising avenue for future research and development.

A potential alternative is the buccal or sublingual route [17,32]. Accessibility and easy administration are the main advantages connected to these types of routes. Furthermore, specific advantages of the buccal delivery route are: (i) it harnesses active (non-passive) mechanisms of buccal absorption such as carrier-mediated facilitated diffusion, active transport, endocytosis, and exocytosis; (ii) the physicochemical properties of peptides can be engineered to make them more suitable for transport via the paracellular pathway; and (iii) penetration enhancers can be used to enhance absorption while adhering to stringent safety criteria to prevent local and systemic toxicities [33]. Moreover, buccal delivery is an oral administration route that achieves direct absorption into the systemic circulation by bypassing the gastrointestinal tract’s stringent conditions, offering certain similarities with specific parenteral administration routes [34]. However, these tablets can create irritation to the gums or risks of choking [35]. Solutions to the side effects of buccal or sublingual tablets rely on the engineering and design of the tablets. A few factors can enhance the absorption of peptides into the buccal or sublingual mucosa. First of all, the encapsulation process and the creation of peptides-in-nanoparticles can greatly enhance the absorption of peptides through the buccal mucosa [17]. By encapsulating peptides, several parameters can be improved: the protection of the drug from degradation and the diffusion rate of the drug across the mucus layer. The most common technique to enhance drug delivery through the buccal mucosa is using hydrophobic polymers since they do not dissolve in saliva. In addition, nanoparticles with a neutral charge or that are positively charged show better adhesion of the buccal mucosa thanks to the negative charges generated by the sialic acid in the mucus [36]. Drug efficiency and efficacy in buccal delivery can be improved by properly engineering the systems, such as tablets or films [37]. For instance, mucoadhesive polymers can enhance the adhesion between the delivery system and the buccal mucosa [38]. Conversely, a water-repellent polymer can be used as a barrier between the delivery system and the oral environment [37].

This article, thus, aims to be a guideline for future investigators in creating buccal or sublingual tablets for the delivery of drugs used to treat diabetes. The review covers the creation of nanoparticles containing peptides, such as insulin or GLP-1 receptors. Later, the production of a system, such as tablets or film, is conceptualized concerning the enhancement of the drug’s efficiency or efficacy. Figure 1 summarizes the aspects described in this review article.

## 2. Peptides for Treating Diabetes

Peptide drugs have emerged as a promising class of therapeutic agents for treating diabetes mellitus due to their high therapeutic efficacy [40,41]. Insulin, consisting of 51 amino acids, plays a crucial role in regulating glucose homeostasis by promoting glucose uptake in insulin-responsive tissues and suppressing glucagon secretion from pancreatic α-cells [42,43]. T1DM and T2DM patients are often both dependent on multiple daily insulin injections to maintain their blood glucose levels within the normal range [44]. There are a few types of insulin, depending on the release rate. For instance, the so-called rapid-acting insulin works in about 15 min, while long-acting insulin can maintain levels of glucose for up to 24 h [45].

Glucagon-like peptide 1 (GLP-1) has also been shown to be an effective drug for T2DM treatment [46,47]. This peptide hormone increases insulin secretion from pancreatic β-cells while inhibiting glucagon release from pancreatic α-cells [48]. Additionally, GLP-1 promotes pancreatic β-cell proliferation and slows down the progression of T2DM. Furthermore, the glucose-dependent effect of GLP-1 allows it to avoid the risk of hypoglycemia [46,47]. However, GLP-1 has a short half-life in blood (<2 min) and is easily degraded by dipeptidyl peptidase-4 (DDP-4), which has limited its clinical application [49].

GLP-1 receptor agonists (GLP-1RAs) have recently been developed to overcome this limitation. These GLP-1RAs, such as exenatide, liraglutide, and semaglutide, have received considerable attention as potential T2DM treatments [50]. Compared to native GLP-1, GLP-1RAs are resistant to DDP-4 degradation and have a longer plasma half-life. These peptides bind to GLP-1 receptors expressed on pancreatic β-cells, promoting glucose-dependent insulin secretion [50]. As well as GLP-1, other peptide drugs have also shown potential as therapeutic agents for diabetes mellitus. For example, amylin, an amyloid peptide co-secreted with insulin, has been shown to improve postprandial glucose control and can reduce insulin usage in T1DM and T2DM patients [51]. Additionally, peptide-based inhibitors of dipeptidyl peptidase-4 (DPP-4) have been developed to enhance the effects of native GLP-1 and other incretin hormones [52]. Apart from these, C-peptide, also known as connecting peptides, also hold special importance in diabetes therapeutics. Co-secreted with insulin from pancreatic β-cells in equimolar amounts, C-peptides reflect the precursor molecule’s cleavage during insulin biosynthesis and have been used historically as a marker for insulin release. New emerging evidence suggests that C-peptide exerts significant bioactive effects on cellular processes critical for glucose homeostasis and vascular function. The consideration of C-peptide as a therapeutic agent holds particular significance in the context of type II diabetes, where insulin resistance and β-cell dysfunction often co-exist. The development of these peptide drugs with improved pharmacological properties and therapeutic efficacy has provided new avenues for the treatment of diabetes mellitus.

More details on the most common peptides used to treat diabetes are listed in Table 1.

## 3. Nanoparticles’ Preparation

The primary reason for the inadequate oral bioavailability of peptide drugs is their low permeability across the epithelium and poor stability in the gastrointestinal environment. Addressing these issues is crucial for improving the bioavailability of orally administered peptide drugs. Researchers have invested significant efforts in developing oral nanoparticles (NPs) capable of encapsulating peptide drugs [58]. The NPs show a major advantage, which is their size. Banerjee et al. [59] compared different diameters of polystyrene NPs and the cell uptake and transport into Caco-2, immortalized cell line of human colorectal adenocarcinoma cells. It was observed that particle uptake by Caco-2 was about 20 and 30% higher for polystyrene NPs with a diameter of 50 nm compared to polystyrene NPs with a 200 and 500 nm diameter, respectively [59]. Overall, researchers confirmed that NPs with diameters below 200 nm are more disposed to penetrating through the epithelium and across the basal membrane [60]. As well as the dimensions, the shape can have an impact on Caco-2 cell uptake. Surprisingly, Caco-2 cell uptakes were significantly higher for rods and discs (~20% after 5 h) with spherical particles (~14% after 5 h) [59]. While dimension and shape are important factors to consider, the materials’ selection is the most important parameter influencing the behavior of encapsulated drugs. The selection of the materials used to form NPs to be orally delivered is very challenging, and it should be carried out by considering the following parameters: the required size of NPs, drug loading, zeta potential, surface characteristics (i.e., charge and shape), association efficiency, biocompatibility and toxicity, the inherent properties of the protein/peptide incorporated (i.e., solubility and stability), the degree of biodegradability, the protein/drug release profile looked-for, and the antigenicity of the final product [61].

These NPs aim to overcome the challenges associated with the gastrointestinal tract and increase the oral bioavailability of peptide drugs. Various materials have been employed to construct oral peptide drug NPs, including lipid NPs and polymer-based NPs. Figure 1 depicts these types of oral peptide drug NPs. The multiple processes used to create NPs are shown in Table 2.

### 3.1. Lipid-Based NPs

Lipid nanoparticles (NPs) are drug delivery systems composed of lipid molecules that can efficiently encapsulate both hydrophilic and hydrophobic drugs [72]. Given that liposomes are known to be impressive in terms of their biocompatibility and drug-loading capacity, they have been the subject of extensive research for encapsulating hydrophilic peptide drugs [73]. Peptides are generally placed within the aqueous layer, where they are protected from enzymatic or gastrointestinal degradation by the outer hydrophobic lipid bilayer. For buccal delivery, neutral liposomes are of special interest due to their hydrophilic and electroneutral surface that permits strong mucoadhesivity. On the other hand, these neutral liposomes are not very effective for gastrointestinal delivery due to their weak interaction with intestinal epithelial cells [74]. Furthermore, due to their bilayer structure akin to the cell membranes, liposomes are reported to be more efficient in delivering peptides to cells [73].

**Glucose-Responsive Oral Insulin Liposomes:** In order to overcome the limitations of neutral liposomes in interacting with intestinal epithelial cells, modifications were made to the surface of neutral liposomes by adding target ligands. An example of this is the glucose-responsive oral insulin liposome, which was reported by Yu et al. for after-meal glycemic regulation [34,75]. Herein, they used a fragment crystallizable (Fc) receptor modification approach, wherein insulin was loaded within a neutral neonatal fragment crystallizable (Fc)-receptor-targeted liposomes core and an outer shell of glucose-responsive phenylboronic acid (PBA)-conjugated hyaluronic acid (HA-PBA). An increase in glucose concentration allowed the detachment of HA shells due to preferential binding of PBA with glucose, enabling the exposed Fc-receptor groups to facilitate insulin transport due to their binding with immunoglobulin G. This resulted in 20.7% encapsulation efficiency, as well as a higher plasma insulin concentration and lower blood glucose in animal studies on diabetes-induced mice [75].

**Protein Corona Liposomes:** In contrast to neutral liposomes, cationic liposomes (CLs) tend to interact with negatively charged mucin, which can facilitate adhesion but may limit their effectiveness in traversing the intestinal epithelial layer [76]. This adhesion is driven by favorable electrostatic interactions between positive and negative charges. It is noteworthy that while electrostatic interactions play a role in adhesion, a balanced charge state is essential to achieve efficient intestinal transport. Hence, to overcome potential challenges arising from excessive positive charge, CLs can be modified by coating them with negatively charged materials, thereby reducing their net positive charge. For instance, Ding et al. developed protein corona liposomes (Pc-CLs) for oral liraglutide delivery [77]. They prepared CLs using distearoylphosphatidylcholine (DSPC), Chol-PEG-AT-1002 by the double-emulsion method. BSA was then adsorbed onto the surface of the CLs to form Pc-CLs. After BSA adsorption, Pc-CLs’ diameter increased from 127 nm to 202 nm, and the zeta potential decreased from +36.1 to 1.76 [77]. Insulin was loaded into the Pc-CLs using electroporation. The encapsulation efficiency and loading capacity of insulin in Pc-CLs were 84.63% and 2.08%, respectively [77]. These Pc-CLs demonstrated a significant improvement in mucus-penetrating velocity compared to native CLs. Enzymes could gradually hydrolyze the BSA protein corona when the Pc-CLs crossed the mucus layer. The exposed CLs could interact with the underlying intestinal epithelium to enhance transepithelial transport. In the in vivo rats study, the in vivo distribution showed that the Pc-CLs group significantly increased the oral bioavailability of liraglutide, which had a longer absorption time than other groups [77].

**Lipid-based micelles:** These are versatile drug delivery systems that offer many advantages over traditional drug delivery methods. One key advantage is their ability to encapsulate drugs in hydrophobic cores, which protects the drug from degradation and clearance by the immune system. Phospholipid micelles comprise a hydrophobic core and a hydrophilic outer shell made of polyethylene glycol (PEG) [78]. The PEG provides steric stabilization and prevents aggregation of amphiphilic peptides. This type of micelle has been shown to induce peptide transition from an unstable conformation to a stable alpha-helical conformation, which is desirable for many peptides, including glucagon peptides [78]. DSPE-PEG2000 is a commonly used phospholipid in self-assembling sterically stabilized micelles (SSM) due to its biocompatibility. Lipid-based micelles can have a size ranging from 100 to 200 nm and have been shown to efficiently deliver peptides and proteins to target cells [79]. Studies have shown that phospholipid micelles can enhance the stability and bioactivity of peptides and proteins. For example, N-octyl-N-arginine chitosan micelles have been used as an oral delivery system for insulin, resulting in an increased uptake rate from the Caco-2 monolayer and improved oral bioavailability [79]. Polystyrene co-maleic acid (SMA) micelles have also been used to encapsulate insulin and efficiently stimulate glucose uptake in hepatic cells, as well as for transport across the intestinal epithelium.

**Zwitterionic lipids such as DLPC MSNs:** Several efforts have been devoted to identifying neutral lipids that could strongly interact with the intestinal epithelial cell layer. One study by Gao et al. developed a straightforward zwitterionic-based delivery system for mesoporous silica nanoparticles (MSNs), which featured a dilauroylphosphatidylcholine (DLPC) self-assembled on the surface of the hydrophobic MSN to form a neutral and hydrophilic coating [80]. The insulin-loaded DLPC MSNs had a size of 164.8 nm and a nearly neutral charge surface with 5.9 mV of zeta potential [80]. The DLPC material, which possesses a hydrophilic zwitterionic PC headgroup and is considered “muco-inert,” enabled mucus permeation. Additionally, the hydrophilic head of DLPC exhibited a strong affinity for the intestinal peptide transporter PEPT 1. Compared to MSNs, the DLPC MSNs significantly enhanced cellular uptake by 2.5-fold [80]. The oral bioavailability of the insulin-loaded DLPC MSNs was able to decrease 55% of the original blood glucose level in STZ-induced type 1 diabetic rats, which was significantly greater than that of free oral insulin and that of unmodified MSNs [80]. This study suggests that zwitterionic lipids may represent a novel oral peptide drug delivery approach in diabetes mellitus treatment.

Despite the potential benefits of using lipid-based nanoparticles (NPs), the lower encapsulation efficiency of peptide drugs in lipid-based NPs and poor stability in biological fluids and during storage results in increased treatment costs. The more recent achievements of lipid-based NPs in the oral delivery of diabetes-treating peptides are listed in Table 3.

### 3.2. Polymer-Based NPs

Researchers have investigated using nanoparticles (NPs) made from both natural and synthetic polymers, or their combinations, for oral peptide drug delivery.

**Natural polymers such as chitosan and their combinations with other natural polymers:** Chitosan (CS), alginate, dextran, gelatin, etc., are examples of natural polymers commonly employed in peptide delivery due to their non-toxicity and biocompatibility [17]. CS, a chitin-derived polysaccharide, has received most attention for peptide delivery due to its ability to increase residence time for peptide delivery by adhering to mucosal and cellular surfaces [84,85]. Furthermore, CS can be easily engineered by changing its molecular weight and degree of acetylation to suit different applications and drug delivery pathways [85]. CS also acts as a permeation enhancer by reversibly opening cell tight junctions and facilitating the transport of peptides via the paracellular pathway and epithelial cell layers [17,18]. The positively charged amino groups on the CS backbone can interact with negatively charged macromolecules, such as integrin α_γ_β_3_ on the cell membrane, leading to the redistribution of claudin-4 (CLDN4) from the cell membrane to the cytosol, where it is reversibly degraded in lysosomes, to increase paracellular permeability [86]. CS can improve the transport of drugs across intestinal mucosal membranes allowing polar drugs to penetrate [87]. For example, insulin-loaded chitosan-ethylenediaminetetraacetic acid hydrogel films displayed a mucoadhesive force over 17,000 N/m^2^, which continued for 4 h in a simulated oral cavity [88]. CS contains free amine groups and can form ionic crosslinkages with multivalent anions. In our own work, Guo et al. [18] developed pH-responsive CS/sodium tripolyphosphate (TPP) crosslinked nanoparticles (NPs) for the oral delivery of insulin. With excessive CS, the NPs had a diameter of around 318 nm and zeta potential of 11 mV [18]. The insulin loading efficiency and content were 98% and 25%, respectively [18]. The NPs could adhere to and penetrate the mucus layer and approach the epithelial cell surface, particularly the intestinal cells (pH 6.0–6.6) [18]. The CS NPs opened the tight epithelial junctions, increasing insulin’s paracellular permeability. The release behaviors of all these dehydrated NPs showed that they could all fast-release in the solution with a pH = 2.5 and a pH = 7, while they were very stable in the solution with a pH of 6.5 [18]. However, they could be disintegrated at a lower pH in the stomach. In vitro cell studies showed that using the optimized insulin NPs can enhance the cell penetration and uptake while having no cell toxicity toward intestinal, liver, and buccal cells [18]. Therefore, the use of CS NPs presents an effective platform for oral peptide drug delivery in the treatment of diabetes mellitus.

Researchers have used CS/alginate NPs to orally deliver the Cp1-11 peptide/insulin complex (CILN), which is reported to improve insulin bioavailability [89]. The use of alginate and CS together led to a small particle diameter (237.2 nm) and higher insulin loading efficiency (90.4%) of CILN [89]. Animal studies on diabetes-induced rats revealed that CILN had a much higher oral bioavailability (15.6%) compared to free insulin (0.1%), demonstrating the potential of CS/alginate NPs as carriers for oral peptide drug delivery [89].

**Biodegradable synthetic polymers and their modifications with PEG and cell-penetrating peptides:** Synthetic polymers such as poly(lactic-co-glycolic acid) (PLGA), poly(ε-caprolactone) (PCL), polyethylene glycol (PEG), and polylactide (PLA) have a more diverse structure and higher mechanical strength than their natural counterparts, making them excellent candidates for oral peptide drug delivery [90]. Researchers have prepared PLGA–PEG NPs for the oral delivery of insulin. Small Ins-NPs can be formed by dissolving insulin and PLGA–PEG molecules in an organic phase (DMSO) with a mean hydrodynamic diameter of 150 nm and an insulin load over 10% and around 90% of conjugation efficiency [91]. The NPs described in this study were decorated with an engineered human albumin variant that had improved human FcRn binding, resulting in a 2-fold increase in transcytosis across polarized epithelia compared with NPs decorated with wild-type albumin [91]. When tested for oral delivery in human FcRn transgenic mice with induced diabetes, these NPs could reduce glycemia by approximately 40% after 1 h, improving pharmacologic availability [91].

In addition, combining natural and synthetic polymers can enhance the oral bioavailability and therapeutic efficacy of peptide drugs in diabetes mellitus. For example, PLGA can achieve a controlled release of peptide drugs and protect them from enzymatic degradation in the GI tract. However, negatively charged PLGA NPs face limited penetration through the mucus and epithelial cell layer. To overcome this limitation, Araújo et al. used microfluidics technology to encapsulate GLP-1-loaded PLGA NPs and a DPP4 inhibitor (iDPP4) with an enteric HPMC-AS (hydroxypropylmethylcellulose acetylsuccinate) polymer to increase intestinal permeation [92]. In STZ-nicotinamide-induced T2DM rats, the dual-delivery H-PLGA particles reduced blood glucose levels by 44% and remained constant for another 4 h after oral administration [92]. This process results in the possibility of using natural and synthetic polymers in the design of oral peptide drug delivery systems with enhanced efficacy in treating diabetes mellitus. The combination of PLGA NPs with CS and CPP improves intestinal permeation, increasing the potential for the successful oral delivery of peptide drugs. These findings have implications for improving the therapeutic options for diabetes and other diseases requiring oral peptide drug delivery.

**Non-biodegradable synthetic polymers such as pHPMA:** poly-N-(2-hydroxypropyl) methacrylamide (pHPMA) is a non-biodegradable hydrophilic synthetic polymer that is frequently utilized in oral peptide drug delivery for diabetes mellitus treatment. pHPMA can serve as a mucus-inert coating material, which facilitates mucus permeation. A study by Liu et al. provided evidence for the potential of using mucus-inert polymer-coated mucoadhesive NPs to improve oral drug delivery. The muco-bioadhesive strength surges with the molecular weight, mucoadhesive polymer concentration, chain flexibility, presence of hydrogen-bond-forming groups (hydroxyl, carboxyl, amines, and amides), positively or negatively charged groups, and reduced crosslinking density [88]. The pHPMA coating allows for excellent mucus permeability and efficient interaction with the underlying epithelial cells, facilitating the paracellular transport of the loaded drug via the opening of tight junctions [93]. In diabetic rats, the developed NPs exhibited a remarkable hypoglycemic response and increased serum insulin concentration after oral administration [93]. They highlight the importance of considering mucosal tissue, including the secreted mucus layers, as complex absorption barriers, and the results demonstrate the feasibility of overcoming these barriers in a multistep process. Furthermore, the extensive use of HPMA polymer as a drug carrier makes this delivery platform a promising candidate for clinical translation.

**Other biologically inspired polymers such as pUDCA:** Polymers derived from monomers with biological activity can also modulate the release of peptide drugs. For example, Lee et al. [94] formulated the ursodeoxycholic acid (UDCA) polymer (pUDCA) into NPs. UDCA is a bile acid monomer that can lower insulin resistance in T2DM, and when used as a protective insulin carrier, it imparted the dual property of being a bile acid receptor agonist to improve insulin absorption. In vivo studies showed that they not only controlled blood glucose levels in T1DM mice and pigs but also reversed inflammation, improved metabolic activity, and resulted in the longer survival of animals [94]. The metabolic and immunomodulatory functions of pUDCA NPs offer translational opportunities to prevent and treat T1DM. In conclusion, polymerized UDCA, and formulating it into NPs, offers a promising strategy for enhancing oral insulin delivery and treating T1DM. The pUDCA NPs serve as insulin carriers and activators of TGR5, which promotes insulin secretion and enhances glucose tolerance. Additionally, the NPs can polarize intestinal macrophages towards the M2 phenotype and reverse inflammation, offering therapeutic potential for preventing and treating T1DM. Further research could focus on optimizing the pUDCA NPs to improve their efficacy and safety for clinical translation.

Polymer-based NPs have a high encapsulation efficiency for peptide drugs, and their diverse chemistry enables them to have multiple functions. However, these NPs are susceptible to disaggregation upon dilution in biological fluids, leading to the early release of peptide drugs. This rapid release of drugs early on in the gastrointestinal (GI) tract, or in the saliva during buccal delivery, can reduce peptide bioavailability and therefore impact the hypoglycemic effect. More recent achievements of polymer-based NPs in the oral delivery of diabetes-treating peptides are listed in Table 4.

## 4. Drying Procedure to Form Dry Powders

Once the nanoparticles are created to protect peptides, suspensions need to be dried before being placed in tablets for buccal or sublingual delivery. Powders are generally more convenient for the patient than gels and ointments. Furthermore, solid systems usually show a higher stability and higher residence time at the buccal mucosa, thus generating a faster mucosal absorption [88]. For these reasons, buccal or sublingual films or tablets are commonly coated or filled with dry powders [102,103]. In order to obtain dry powders, two techniques can be defined as the most common, freeze and spray drying, as shown in Figure 2. A few literature references have proposed an extensive comparison between the two techniques [104]. Freeze drying is simpler, and it has a high yield [18]. In contrast, spray drying shows a lower yield. However, spray drying can avoid aggregation changes during the drying process [105,106]. Moreover, it shows much faster drying times.

**Freeze Drying:** Most of the previous references involving the production of nanoparticles containing a peptide for diabetes treatment selected freeze drying as the drying procedure, as shown in Table 5 [88,107]. The main reasons for this trend are the high availability and low costs of freeze drying. However, there are some drawbacks, such as the deterioration of some materials chosen for nanoencapsulation by freezing temperatures [108]. To this purpose, the use of cryoprotectants is recommended. Fonte et al. [107] suggested using sucrose, trehalose, or glucose to protect insulin-loaded NPs during freeze drying. Cryoprotectants are confirmed to stabilize the NPs during the drying steps due to the generation of hydrogen bonds between the cryoprotectant and the polar groups at the surface of the NPs [109]. The strong bond between the cryoprotectant and the surface of insulin-loaded NPs makes the release of insulin harder, thus protecting it during the drying process. Fonte et al. demonstrated that glucose (10% *w*/*w*)-added NPs released 30% in the first 2 h, which is less than the 45% with respect to the non-added cryoprotectant sample [107]. Cryoprotectants are not necessary in the spray drying procedure. Here, the dry powder is formed following the theory of particle engineering. A falling droplet exposed to a heat flow and travelling through a cyclone generated microparticles with morphological properties dependent on the spraying conditions [110,111]. In this drying procedure, exposure to freezing temperatures is not required. However, insulin-loaded NPs might be exposed to high temperatures for a short period of time.

**Spray Drying:** Our previous work demonstrated poor insulin bioavailability damage induced by the spray drying procedure [18]. For instance, Guo et al. [18] used 100 °C for spray drying insulin-loaded NPs and indicated only a loss of insulin of 5% compared to fresh samples. Guo et al. [18] also showed that introducing only 1% *w*/*w* of mannitol highly increased the spraying yields but jeopardized the stability of insulin with over 30% of bioavailable insulin lost during such a spray drying procedure [18]. A motivation for having an additional component in either freeze or spraying drying is to reduce the adhesion forces between NPs and, thus, the formation of aggregates. Some polymers can be selected, such as hydroxypropyl methylcellulose (HPMC), see Table 5. HPMC has been used for mucoadhesion properties, which enhance the accuracy of the location of deposition [38]. However, the addition of any material in the drying procedure can modify the size of the NPs. The higher the material quantity added, the larger the NPs can grow in spray drying [112]. In freeze drying, quantities lower than 10% *w*/*w* are suggested; higher quantities can interfere with the homogeneity of the release rate of insulin [18].

**Table 5 pharmaceutics-15-02265-t005:** List of the most cited references on the generation of dry powders for the sublingual or buccal delivery of GLP-1 peptides.

GLP-1	Drying Techniques	Drying Parameters	Additives	Ref.
Human insulin	Freeze	NA	NA	[88]
Linagliptin and empagliflozin	Vacuum	60 °C and 5 mbar	Pluronic F68, Pluronic F-127, fluorophore umbelliferon and eosine	[113]
l-Cysteine	Freeze	−30 °C at 0.01 mbar	NA	[33]
D,l-valine	Vacuum	25 °C and 1000 mbar	Eudragit (ERL) and hydroxypropyl methylcellulose (HPMC)	[102]
Human insulin	Freeze	NA	NA	[114]
Human insulin	Freeze	−60 °C and 0.09 mbar	10% *w*/*w* fructose, trehalose, or glucose	[107]
Human insulin	Spray	100 °C, feeding flow of 3 L/min, and airflow of 4 L/min	Mannitol (1% *w*/*w*)	[18]
Human insulin	Freeze	0 °C at 0.133 mbar	NA	[115]

Freeze and spray drying are only two of the possibilities for forming dry powders composed of peptide-loaded NPs. One of the most recent efforts to create a system for delivering insulin-loaded NPs was drying the suspension in a 3D printing device [116]. Here, a combination of sodium alginate/polyethylene glycol was used to create scaffolds involving different weight percentages of dry insulin-loaded NPs [116]. Microfluidics, as discussed in the case of a dual-delivery system encapsulating GLP-1-loaded PLGA NPs and DPP4 inhibitor (iDPP4) within an enteric HPMC-AS polymer through a microfluidics system, have also been used as a possible approach of formulation [92].

## 5. Concept for Buccal Tablets for the Buccal Delivery of Nanoparticles Containing Insulin, GLP-1, and their Analogs

The most common delivery of dry powders containing NPs of encapsulated peptides is through some typical oral delivery systems, such as oral films, capsules, pills, or syrups. These systems are known for the overall efficacy of the carried protein/peptide and can be modified to target specific tissues or organs. A few techniques exist for the delivery of peptides to the buccal or subbuccal areas [37]. The presented strategy of buccal dosage forms, such as patches, films, and tablets, involves:Single-layer devices from which the drug is released multidirectionally;Devices including an upper water-resistant layer that reduces the wastage of the drug into the oral cavity and avoids the degradation of drug by salivary enzymes;Unidirectional release devices, from which drug loss is low since the drug is released only from one side, facing the buccal mucosa [35].

The most common technique used for delivering peptides to the buccal or sublingual areas is that of tablets, normally in the shape of 5–8 mm diameter compressed disks. Buccal tablets have been the most commercially available dosage form for buccal mucosal delivery; however, the main drawback of this technique is the lack of physical flexibility, which has led to poor patient compliance for long-term and repeated doses. Therefore, much effort has been made to modify the traditional form of tablets [117]. Films are another very common form of delivery, as shown in Figure 3. Other terms used in the literature can be associated with films; such examples are patches or dressings. As opposed to tablets, films are flexible, and thin dosage forms can be contained in a large portion of a film-forming polymer that is adequately plasticized. The small thickness of films can be a disadvantage; the dose used in films is definitely lower compared to that in tablets. However, previous references describe that about 50% of the drug loaded into the film can be released into the bloodstream [118]. Other systems for the delivery of dry powders of encapsulated peptides can be wafers or lozenges. Wafers or xerogels are solid mucoadhesive dosage models regularly engineered through lyophilization to eliminate the solvent and form a solid porous structure. The high porosity allows adequate contact and drug release, which can also be designed using mucoadhesive polymers [119]. Lozenges are compressed dose models designed to gradually dissolve and release drugs through the patient’s saliva flow and tongue movement effort. Their design allows the dissolution of drugs between 10 and 30 min. The main drawback of this delivery form is the consideration of their taste and the high loss of drugs [120].

Herein, we present the concept of triple-layered buccal mucoadhesive tablets and films, as shown in Figure 3, with small tablets (less than 1cm in diameter) being the preferred formulation, as films need to be removed after the drug elution is complete while tablets can be swallowed. In general, these tablets are composed of different layers to facilitate the mucosa’s adhesion and reduce the drug’s dissolution into the saliva. Therefore, they must contain a layer including the active compound, a layer composed of a mucoadhesive polymer, and a layer consisting of a water-repellent polymer. The combination of these three layers in different forms, as well as their small size, can allow these tablets to have a high residence time in the buccal or sublingual area. Therefore, multilayered methods are preferred for unidirectional release concerning the mucosa and enhanced absorption. When using mucoadhesive polymers, such tablets can remain in contact for minutes up to a few hours.

As shown in Figure 3, mucoadhesivity is possibly the most critical factor in improving the release and delivery of peptides. Some of the materials used for their mucoadhesive properties are listed in Table 6. When approaching the design of a buccal or sublingual tablet or film, the accountable properties of the layer facing the mucosa are mucoadhesivity, water solubility, tensile strength, and viscosity, as well as the highest mucoadhesion and the lowest chance of a detachment between the tablet or film and the buccal or sublingual mucosa. Thus, hydroxyethyl cellulose, chitosan, or carbopol can be positive candidates. However, for instance, carbopol is highly soluble in water; thus, part of the encapsulated peptides could be lost in the saliva [18]. The parameters of tensile strength and viscosity relate to the patient’s comfort. A high tensile strength shows a high hardness of the tablet and a possible generation of some sort of irritation on the tissues of the patients. Low viscosity can lead to a weak attachment between the tablet or film and the buccal or sublingual mucosa [37]. In addition to the selections shown in Table 6, the existence of thiolated polymers is worth mentioning. Several previous references emphasize the high adhesion forces between thiolated polymers and different types of mucosa. However, the mucoadhesion greatly depends on the swelling and crosslinking of the polymer [121]. Another trick used by previous researchers is the use of permeation enhancers, which can be combined in the formulation. However, the enhancer’s wetting and quick release must be adjusted for a concomitant drug permeation [119].

As for the other side of the tablets, hydrophobic polymers are commonly selected. Some examples are shown in Table 6 and the optimal candidates are carboxymethyl cellulose, hemicellulose, or polydimethylsiloxane. However, poor solubility in water makes the dissolution of the tablet in saliva impossible, even after several hours of application. The water-repellent layer must be hard and poorly soluble in water to stand the environment in the patient’s mouth. Thus, carboxymethyl cellulose could possibly be the best selection among those shown in Table 6.

## 6. Conclusions and Future Developments

The buccal or sublingual delivery of peptides aimed at curing diabetes has shown positive results in several recent research studies. Such an appeal is due to the high accessibility and high bioavailability of the drug, and its high ease of use. However, the effect of buccal and sublingual tablets loaded with peptides for treating diabetes can vary greatly according to several parameters. Firstly, the selection of the peptide to use can vary from the traditional insulin to more recently used peptides, such as the GLP-1RAs and other peptides. The latter are less studied with respect to insulin; however, they can be applied to other conditions as well as diabetes. Therefore, future investigations are expected to dig more into the generation of buccal or sublingual tablets containing GLP-1RAs. Either using insulin or GLP-1RAs, a nanoencapsulation procedure is highly recommended. In fact, several references have emphasized the positive impact of nanocapsules on the drug release rate, half shelf life, and efficacy. Encapsulation with suitable polymers appears to be the most viable and efficient method for creating nanoparticles smaller than 200 nm for delivery into the buccal mucosa. The selection of mucoadhesive and hydrophobic polymers is connected to increased drug bioavailability. Once the nanoparticles are carefully designed, they are usually suspended in a liquid form. Dry forms of nanocapsules have shown a much longer half shelf life, and thus, a stronger stability for several types of delivery. A few drying techniques are available, among which freeze drying is the most popular due to its high accessibility. However, spray drying has been shown to preserve the most encapsulated peptides. Moreover, by introducing additional materials, such as sugars, nanoparticle agglomeration related issues can be avoided. Any agglomeration can greatly impact the release pattern of encapsulated peptides. 

However, dry powders of encapsulated peptides need to be placed in a more complex system in order to be delivered to the buccal or sublingual mucosa. Among the several systems present in the literature, tablets and films are the most common. In particular, unidirectional tablets can increase the efficacy of encapsulated peptides. The design of such a system involves a mucoadhesive layer, attached to the mucosa layer, a layer loaded with an encapsulated drug, and a water-repellent layer, facing the opposite side of the buccal or sublingual mucosa. This design allows the system to be stable in the mouth of the patient and delivers the loaded peptides to the systemic circulation through buccal tissues, while avoiding any loss to unintentional salivary ingestion. Further, such formulations are expected to be shelf-stable and not require refrigerated storage, and thus improve the global outreach of peptide drugs through a more environmentally friendly supply chain network. Further, the elimination of needles and plastic wastes further improves the sustainable aspect of such a system. Last but not the least, the improvement to the quality of life of patients by not requiring daily injections, as well not worry about travel logistics related to refrigerated storage, are principal motivations for the adoption of these approaches.

## 7. Patents

The concept has been patented in the form of a triple-layer buccal mucoadhesive tablet containing insulin nanoparticles designed by us that can rapidly deliver insulin akin to injected insulin, but with a more prolonged blood glucose reduction effect. An in vivo study in rats indicated that our insulin buccal tablets can significantly decrease the blood glucose level compared with the oral administration of free insulin. The results showed that the buccal tablets containing insulin nanoparticles had a similar fast onset of action as that of an i.p. injection, while it can be administrated more conveniently [151]. A WorldWide PCT Application Serial No. PCT/CA2023/051129 was filed 25 August 2023 entitled “MUCOADHESIVE TABLETS WITH UNIDIRECTIONAL RELEASE BEHAVIOR FOR RELEASE OF PEPTIDE THERAPEUTIC PARTICLES”, which claims priority from U.S. Provisional Patent Application Serial No. 63/400,863 filed 25 August 2022 entitled “MUCOADHESIVE BUCCAL TABLETS WITH UNIDIRECTIONAL RELEASE BEHAVIOR CONTAINING INSULIN PARTICLES”. These filings cover the technological aspects of the buccal tablets’ design and MNA-TG-chitosan synthesis.

## Figures and Tables

**Figure 1 pharmaceutics-15-02265-f001:**
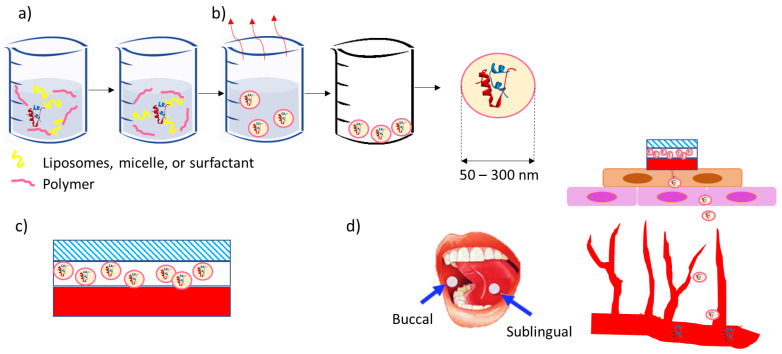
Generation of sublingual or buccal tablets or films loaded with encapsulated peptides aimed to treat diabetes. Some of the pictures have been rearranged and derived from previous references with permission [39]. (**a**) The creation of these tablets or films involves a chemical approach to generate a dispersion of peptide-loaded nanoparticles. (**b**) The liquid part can be removed by a drying procedure to obtain dry powders of peptide-loaded nanoparticles, in diameter ranging between 50 and 300 nm. (**c**) Tablets or films for buccal or sublingual delivery need to be engineered and designed for the optimal release rate and efficiency when reaching (**d**) the capillary system in the patient’s mouth.

**Figure 2 pharmaceutics-15-02265-f002:**
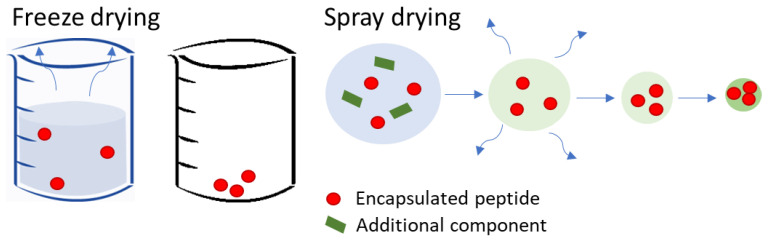
Comparison of the two main drying techniques, freeze drying and spray drying, for the formation of dry powder of encapsulated peptides for the treatment of diabetes. The red spheres represent insulin-loaded nanoparticles, while the green rectangles represent any additional component.

**Figure 3 pharmaceutics-15-02265-f003:**
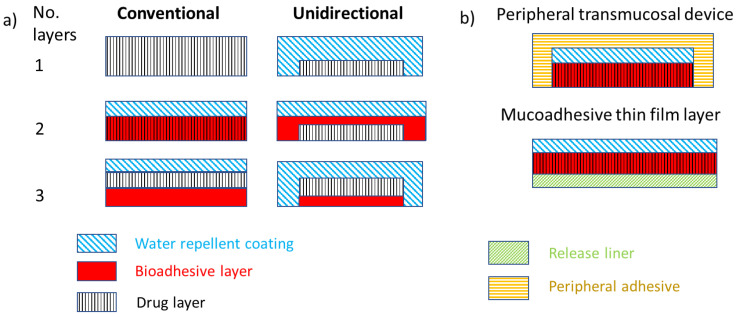
Illustration of the concept of mucoadhesive unidirectional release buccal tablets: (**a**) and films, (**b**) for the buccal or sublingual delivery of peptides to cure diabetes.

**Table 1 pharmaceutics-15-02265-t001:** Properties of the main peptides used for curing diabetes. The solubility is shown for a temperature of 25 °C and a pH of 7.4.

Peptide	Chemical Structure	Detailed Type	Molecular Weight (kDa)	Water Solubility (mg/mL)	Ref.
Short-acting insulin	C_257_H_383_N_65_O_77_S_6_	Humulin R U-100 Novolin R FlexPen ReliOn	5.8	Insoluble	[53]
Rapid-acting insulin	C_257_H_383_N_65_O_77_S_6_	inhaled insulin insulin aspart insulin glulisine insulin lispro	5.8	Insoluble	[53]
Intermediate-acting insulin	C_257_H_383_N_65_O_77_S_6_	insulin isophane	5.8	Insoluble	[54]
Long-acting insulin	C_274_H_411_N_65_O_81_S_6_	insulin degludec	5.8	10^4^	[54]
	C_267_H_402_O_76_N_64_S_6_	insulin detemir	5.9	14.2	
	C_267_H_404_N_72_O_78_S_6_	insulin glargine	6.1	3.6	
Exenatide	C_184_H_282_N_50_O_60_S	GLP-1 stimulates glucose	4.2	3	[55]
Liraglutide	C_172_H_265_N_43_O_51_	GLP-1 stimulates glucose	3.7	270	[56]
Semaglutide	C_187_H_291_N_45_O_59_	GLP-1 stimulates glucose	4.1	1	[57]

**Table 2 pharmaceutics-15-02265-t002:** Advantages or disadvantages of the most common methods used to prepare nanoparticles containing drugs used for the treatment of diabetes. The following references have been used [62,63,64,65,66,67,68,69,70,71].

Preparation Method	Advantages	Disadvantages
High-pressure homogenization (HPP)	Suitable for a wide range of nanocarriers Easily scaled up NPs’ size lower than 300 nm	NPs’ dimensions are influenced by the intensity and duration of energy (homogenization pressure and cycles) input Potential drug degradation for heat-sensitive drugs Risk of drug leaching into the aqueous phase Metallic contamination in the final product
Ultrasonication	Lower energy consumption compared to HPP NPs’ size lower than 200 nm	Low entrapment efficiency (<75%) Potential risk of metallic contamination in the final product
Ion gelation	Small particle sizes (<300 nm) while being energy-efficient	Limited selection of crosslinker and emulsifiers pH shows a strong influence on the results
Phase inversion temperature (PIT)	Small particle size (<200 nm) and energy efficiency	Limited selection of oils and emulsifiers High concentrations of surfactants are needed (>10%) Not suitable for heat-sensitive drugs
Coacervation	Easy scalability of the process Organic solvent-free method	May not be suitable for pH-sensitive drugs
Supercritical fluid method	Solvent-free Products in the form of dry powders	Process can be costly to implement
Solvent emulsification/evaporation	Mild conditions, such as ambient temperature and pressure. Small particle sizes (<400 nm)	Organic solvents are required
Film hydration	Easily scaled up and suitable for larger production volumes Enhanced solubility to desired level depending on the selected solvent	Use of organic solvents Heterogeneity in terms of both size and shape of NPs
Solvent injection	Efficient manufacturing Liposomes with a narrow size (around 200 nm) when ethanol is used	Use of organic solvents
Nano spray drying	Single-step process Cost-effective Good dissolution profile	May not be suitable for heat-sensitive drugs

**Table 3 pharmaceutics-15-02265-t003:** List of the most recent references developing lipid-based nanoparticles for the delivery of drugs used for the treatment of diabetes.

Drug	Encapsulation Method	In Vivo Method/Dose	Delivery Mechanism	Ref.
Insulin	Fc-receptor (FcRn)-targeted liposomes with glucose-sensitive hyaluronic acid (HA) shell	Chemically induced type 1 diabetic mice/10U/kg	The detachment of HA occurs when glucose binds with phenylboronic acid during high postprandial glucose levels. Fc facilitates the absorption of liposomes in the intestines.	[75]
Exendin-4	Chondroitin sulfate-g-glycocholic acid (GCA)-coated liposomes (EL-CSG)	High-fat diet STZ-induced T2DM rats/300 μg/kg	GCA promoted the transportation of liposomes through the layer of intestinal epithelial cells.	[79]
Insulin	Solid lipid nanoparticles (SLNs) modified with stearic acid–octaarginine (SA-R_8_)	Fasted diabetic rats/25 U/kg	Insulin was partially protected from gastrointestinal enzymes by incorporation into SLNs and SA-R_8_.	[81]
Insulin	“Oil-soluble” reversed lipid nanoparticles (ORLN) coated with phospholipid (PC) shell	Fasted diabetic rats/60 μg/kg	The decrease in enzymatic degradation of insulin in the intestinal tract, as well as the increase in drug transcytosis across the intestinal epithelia.	[82]
Insulin	Cell-penetrating peptides (CPPs)—incorporated insulin-loaded solid lipid nanoparticles	Fasted diabetic rats/10 U/kg	Intermolecular interactions between INS and L- and/or D-penetration.	[83]

**Table 4 pharmaceutics-15-02265-t004:** List of the most recent references developing polymer-based nanoparticles for the delivery of drugs used for the treatment of diabetes.

Drug	Encapsulation Method	In Vivo Method/Dose	Delivery Mechanism	Ref.
Exendin-4	Tannic acid/exendin-4/Fe^3+^ ternary nanoparticle system	T2D mice/6 mg/kg	The ternary NPs release exendin-4 in a sustained manner because of pH-induced dissociation after intraperitoneal administration	[95]
Insulin	Hydroxypropyl methylcellulose phthalate (HP55)-coated capsule containing PLGA/RS NPs	Fasted diabetic rats/50 U/kg	Selectively released insulin from NPs in the intestinal tract, instead of stomach, enhances the penetration of insulin across the mucosal surface in the intestine.	[95]
liraglutide	PLA NPs modified with a cyclic, polyarginine-rich, cell-penetrating peptide (cyclic R9-CPP)	Fasted diabetic rats/0.5 MBq	These NPs facilitated the intestinal retention and mucosal penetration of peptide therapeutics.	[96]
Insulin	Positively charged chitosan-coated PLGA NP (CS-PLGA NP)	Fasted diabetic rats/15 U/kg	CS-PLGA NPs show positive charge, mucosal adhesion, and absorption promotion, and thus, a long residence of insulin	[97]
Insulin	PEG-coated silica nanoparticles (SiNP–PEG)	Fasted diabetic rats/15 U/kg	SiNP shows high porosity allowing efficient drug loading	[98]
Insulin	Mesoporous silica NPs (MSNs) modified with a hydrophilic block polymer PLA–PEG	Fasted diabetic rats/80 U/kg	MSNs can decrease hydrophobic forces and achieve mucus-inert or penetrating characteristics.	[99]
liraglutide	Liraglutide/tannic acid (TA)/Al^3+^ NP system based on hydrogen bond formation between liraglutide and TA and stabilized by complex coordination interaction between TA and Al^3+^	Fasted diabetic mice/2 mg/kg	Under physiological conditions (Ph 7.4), the partial ionization of phenol groups weaken the hydrogen bonding, and thus trigger decomplexation and release of liraglutide.	[100]
liraglutide	NPs composed of chitosan and poly-N-(2-hydroxypropyl) methacrylamide (Phpma)	Fasted diabetic mice/5 mg/kg	Chitosan can open the connection of epithelial cells while the water solubility of Phpma helps to penetrate the mucus layer	[101]

**Table 6 pharmaceutics-15-02265-t006:** List of the most common mucoadhesive and water-repellent polymers used to create buccal and sublingual tablets and films. The references related to the properties of the materials are the following [106,122,123,124,125,126,127,128,129,130,131,132,133,134,135,136,137,138,139]. The column labeled “ref” identifies the previous literature, including creating a tablet or a film with the related material. The water solubility is expressed for the temperature of 25 °C and a pH of 7.4. The term viscosity refers to dynamic viscosity.

**Bioadhesive Layer**					
**Polymer**	**Mucoadhesivity (N)**	**Water Solubility (mg/mL)**	**Tensile Strength (MPa)**	**Viscosity (Pa s)**	**Ref.**
Carbopol (940)	6–7	100	0.1–114	3–5	[140]
Sodium alginate	6–9	0.4	30–70	0.25–0.60	[116]
Hydroxyethyl cellulose	25–35	9	1.3 ± 0.7	0.1–8.5	[141]
Hydroxypropyl methylcellulose	1.5–9	10	10–38	0.1–10	[102]
Hyaluronic acid	0.4–5	5	5–50	<10^−4^	[142]
Chitosan	9–11	8	100–130	0.1–1	[143]
Polyvinyl pyrrolidone	4–9	100	0.5–1.4	1.2–1.7	[113]
Polyacrylic acid	0.1–0.5	2–10	0.1–5	0.1–2	[144]
**Water-Repellent Layer**					
**Polymer**	**Hydrophobicity (°)**	**Water Solubility (mg/mL)**	**Tensile Strength (MPa)**	**Viscosity (Pa s)**	**Ref.**
Polyvinylchloride	73–81	10	40–50	0.01–0.05	[145]
Polydimethylsiloxane	88–95	Insoluble	3.1–5.5	4–10	[146]
Hydroxypropyl methylcellulose	57–84	1–500	40–160	4–10	[147]
Hemicellulose	80–90	Insoluble	30–40	0.4–0.45	[148]
Ethylcellulose	73–75	Poor	170–240	0.18–1.10	[145]
Poly(e-caprolactone) (PCL)	40–80	1–3	100–150	800–1300	[149]
Carboxymethyl cellulose	85–120	10	42–47	800–1200	[150]

## Data Availability

Available from Corresponding Author on Reasonable Request.

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
