# Peer review of "Concept for a Unidirectional Release Mucoadhesive Buccal Tablet for Oral Delivery of Antidiabetic Peptide Drugs Such as Insulin, Glucagon-like Peptide 1 (GLP-1), and their Analogs"

_pharmaceutics, 2023, doi:10.3390/pharmaceutics15092265_

Round 1

Reviewer 1 Report

Manuscript “Concept for a Unidirectional Release Buccal Tablet for delivery  of Insulin, and glucagon-like peptide 1 (GLP-1) and its’ analogs: Towards oral delivery of antidiabetic peptide drugs” presented by Pratap-Singh et al. describes a variety of approaches for  delivering peptide drugs into the animal organism. A concept utilizing some of these approaches and allowing more efficient and effective drug delivery is presented. This review covers described topics well with strong experimental, theoretical, and technological backgrounds. A number of tables conveniently summarize different sets of information. Most of the comments placed below are suggestions for improvement of the review.

Title

        Buccal and oral indicate the same administration path. Also, oral and buccal drug administration approaches can be described as separate. “Concept” and “towards” have similarities in meaning as well. The title can be simplified.

Abstract

       Is this true statement taking in account that drug injections also can be considered as a vehicle - “show the highest bioavailability as compared to sprays and other delivery vehicles.”?

       Mentioning some factors that influence drug bioavailability in discussed cases can be useful. Why better bioavailability can be achieved?

Introduction

       Typo in three sentences including this section “the main receptors available are exenatide, liraglutide, semaglutide, albiglutide, lixisenatide, dulaglutide, and beinaglutide. These GLP-1 recep”. Check for use of “GLP-1 receptors” and “injection” and similar. It looks like the same typo present in other places as well.

       Typo in “diffusing capacity of lungs for carbon”.

       Clarify withdrawal in “easy administration and withdrawal”.

       This sounds like a mix of not well-logically linked characteristics especially in (iii) - “Furthermore, specific advantages of the buccal route are (i) the presence of nonpassive mechanisms of buccal absorption, such as carrier-mediated transport and endocytosis; (ii) the physicochemical properties of peptides that may support their carriage via the paracellular pathway; (iii) the mechanism of action of penetration enhancers, and the criteria of denying local and systemic toxicities [36].”. Endocytosis of what? Should be it followed by exocytosis for receptor agonists?

       Define “cellular permeability” used on a number of occasions. Does the term relate to endocytosis and exocytosis or only endocytosis?

Main article’s body

       There is a logical problem. Text before this sentence emphasizes the positive role of the interaction of positive and negative charges in mucus - “In contrast to neutral liposomes, cationic liposomes 245 (CLs) tend to be trapped by negatively charged mucin, which can limit their effectiveness 246 in crossing the intestinal epithelial layer [79]. To overcome this challenge, CLs can be 247 coated with negatively charged materials to reduce their positive charges.”. Some clarifications should be made in all related cases.

       Logical link between the described concept of buccal peptide drug delivery and nanoparticles/liposomes/micelles is not well developed throughout the section. Many discussed developments relate oral deliveries with the digestive tract drug absorption.

       Figure 2. Green triangles are not depicted as triangles.

       Not sure what is meaning of this "insulin NPs can optimize the cell penetration and uptake".

       Not clear use of terminology: “rapid release of drugs in the gastrointestinal (GI) tract can reduce their oral bioavailability”.

       Correct this “shown in Figure 3Error! Reference source not found..”.

       1st sentence of the 7. Patents section is not clear. Injection is mentioned. It looks like rephrasing is needed.

Small adjustments can be made.

Author Response

Manuscript “Concept for a Unidirectional Release Buccal Tablet for delivery  of Insulin, and glucagon-like peptide 1 (GLP-1) and its’ analogs: Towards oral delivery of antidiabetic peptide drugs” presented by Pratap-Singh et al. describes a variety of approaches for delivering peptide drugs into the animal organism. A concept utilizing some of these approaches and allowing more efficient and effective drug delivery is presented. This review covers described topics well with strong experimental, theoretical, and technological backgrounds. A number of tables conveniently summarize different sets of information. Most of the comments placed below are suggestions for improvement of the review.

Title

•        Buccal and oral indicate the same administration path. Also, oral and buccal drug administration approaches can be described as separate. “Concept” and “towards” have similarities in meaning as well. The title can be simplified.

AUTHOR RESPONSE: Thanks for the comments, the title has been simplified, yet both Buccal and Oral are retained. Buccal is a subset of Oral delivery, so our tablets are specifically for Buccal, but it is indeed for Oral delivery. NEW TITLE: "Concept for a Unidirectional Release Buccal Tablet for oral delivery of antidiabetic peptide drugs like insulin, glucagon-like peptide 1 (GLP-1) and its’ analogs"

Abstract

•       Is this true statement taking in account that drug injections also can be considered as a vehicle - “show the highest bioavailability as compared to sprays and other delivery vehicles.”?

AUTHOR RESPONSE:  Specified Buccal Delivery Vehicles: "Furthermore, unidirectional films and tablets seem to show the highest bioavailability as compared to sprays and other buccal delivery vehicles. "

•       Mentioning some factors that influence drug bioavailability in discussed cases can be useful. Why better bioavailability can be achieved?

AUTHOR RESPONSE:  Specified that they are better as they prevent loss to saliva and involuntary GI digestion: "Furthermore, unidirectional films and tablets seem to show the highest bioavailability as compared to sprays and other buccal delivery vehicles. This advantageous attribute can be attributed to their capability to mitigate the impact of saliva and inadvertent gastro-intestinal enzymatic digestion, thereby minimizing drug loss. This is especially pertinent as these formulations ensure a more directed drug delivery trajectory, leading to heightened therapeutic outcomes. "

Introduction

•       Typo in three sentences including this section “the main receptors available are exenatide, liraglutide, semaglutide, albiglutide, lixisenatide, dulaglutide, and beinaglutide. These GLP-1 recep”. Check for use of “GLP-1 receptors” and “injection” and similar. It looks like the same typo present in other places as well.

AUTHOR RESPONSE:  Sorry, this comment is not clear. Perhaps the reviewer refers to the appearance of '-' on word wrapping, which is a manifestation of the recommended MDPI font style. I think it will be okay when it goes to print. We have checked by adding some spaces before the word, and then the hyphen disappears...

•       Typo in “diffusing capacity of lungs for carbon”.

AUTHOR RESPONSE:  Corrected to "diffusion capacity of lungs" 

•       Clarify withdrawal in “easy administration and withdrawal”.

AUTHOR RESPONSE: DELETED 'and withdrawal', as it was awkward!

•       This sounds like a mix of not well-logically linked characteristics especially in (iii) - “Furthermore, specific advantages of the buccal route are (i) the presence of nonpassive mechanisms of buccal absorption, such as carrier-mediated transport and endocytosis; (ii) the physicochemical properties of peptides that may support their carriage via the paracellular pathway; (iii) the mechanism of action of penetration enhancers, and the criteria of denying local and systemic toxicities [36].”. Endocytosis of what? Should be it followed by exocytosis for receptor agonists?

AUTHOR RESPONSE: Actually Langoth et al. [36] have described and we merely summarized, which is why it may seem non-logically linked. That is also why we chose to present it in distinct bullet points. Endocytosis refers to the endocytosis of the peptide molecules. Exocytosis for receptor agonists might also happen. Obviously endocytosis and exocytosis happen in the same system, but for purpose of drug delivery they are often represented as a NET (Endocytosis - Exocytosis) effect. In light of your valuable comments, we have revised the paragraph to make it more logical and cover exocytosis  "(i) it harnesses active (non-passive) mechanisms of buccal absorption like carrier-mediated facilitated diffusion, active transport, endocytosis and exocytosis; (ii) the physicochemical properties of peptides can be engineered to make them more suitable for transport via the paracellular pathway; (iii) penetration enhancers can be used to enhance absorption, while adhering to stringent safety criteria to prevent local and systemic toxicities [36]. "

•       Define “cellular permeability” used on a number of occasions. Does the term relate to endocytosis and exocytosis or only endocytosis?

AUTHOR RESPONSE: Cellular permeability refers to to the combined influence of a combination of passive diffusive, facilitated diffusion, active transport, endocytosis and exocytosis. However, we did not use the term cellular permability at all, and we could not find a place to define it. This term is well defined in literature.

Main article’s body

•       There is a logical problem. Text before this sentence emphasizes the positive role of the interaction of positive and negative charges in mucus - “In contrast to neutral liposomes, cationic liposomes 245 (CLs) tend to be trapped by negatively charged mucin, which can limit their effectiveness 246 in crossing the intestinal epithelial layer [79]. To overcome this challenge, CLs can be 247 coated with negatively charged materials to reduce their positive charges.”. Some clarifications should be made in all related cases.

AUTHOR RESPONSE: thanks so much for the insightful comment. We have now modified to make the context of charges much clearer:  "In contrast to neutral liposomes, cationic liposomes (CLs) tend to interact with negatively charged mucin, which can facilitate adhesion but may limit their effectiveness in traversing the intestinal epithelial layer [79]. This adhesion is driven by favorable electrostatic interactions between positive and negative charges. It's noteworthy that while electrostatic interactions play a role in adhesion, a balanced charge state is essential to achieve efficient intestinal transport. Hence, to overcome potential challenges arising from excessive positive charge, CLs can be modified by coating them with negatively charged materials, thereby reducing their net positive charge. For instance, Ding et al. developed protein corona liposomes (Pc-CLs) for oral liraglutide delivery [80]. "

•       Figure 2. Green triangles are not depicted as triangles.

AUTHOR RESPONSE: Sorry, it should have been green rectangles. It was corrected in Figure 2 Caption.

•       Not sure what is meaning of this "insulin NPs can optimize the cell penetration and uptake".

AUTHOR RESPONSE: Replaced optimize with enhance for better clarity: "In vitro cell studies showed that using the optimized insulin NPs can enhance the cell penetration and uptake while having no cell toxicity toward intestinal, liver, and buccal cells [18]. "

•       Not clear use of terminology: “rapid release of drugs in the gastrointestinal (GI) tract can reduce their oral bioavailability”.

AUTHOR RESPONSE: Corrected for clarity to "This rapid release of drugs early on in the gastrointestinal (GI) tract, or in the saliva during buccal delivery, can reduce peptide bioavailability and therefore impact the hypoglycemic effect. "

•       Correct this “shown in Figure 3Error! Reference source not found..”.

AUTHOR RESPONSE: Corrected.

•       1st sentence of the 7. Patents section is not clear. Injection is mentioned. It looks like rephrasing is needed.

AUTHOR RESPONSE: Corrected to: "The concept has been patented in the form of a triple layer buccal mucoadhesive tablets containing insulin nanoparticles designed by us that can rapidly deliver insulin akin to injected insulin, but with a more prolonged blood glucose reduction effect. "

Reviewer 2 Report

This communication review manuscript aims to be a guideline for future investigations in creating buccal or sublingual tablets for the delivery of drugs used to treat diabetes. Specifically, the review covers the creation of nanoparticles containing peptides, such as insulin or GLP-1 receptors.  Their chemical encapsulation via liposomes or polymers is suggested as the most viable and efficient method for creating nanoparticles smaller than 200 nm for delivery into the buccal mucosa. The selection of mucoadhesive and hydrophobic polymers is connected to increased drug bioavailability.

The authors conclude that among the several systems present in the literature, tablets and films are the most common. In particular, unidirectional films can increase the efficacy of encapsulated peptides. 

The manuscript is concisely written, well documented and of interest to both the cognizant and non-cognizant reader.

Author Response

This communication review manuscript aims to be a guideline for future investigations in creating buccal or sublingual tablets for the delivery of drugs used to treat diabetes. Specifically, the review covers the creation of nanoparticles containing peptides, such as insulin or GLP-1 receptors.  Their chemical encapsulation via liposomes or polymers is suggested as the most viable and efficient method for creating nanoparticles smaller than 200 nm for delivery into the buccal mucosa. The selection of mucoadhesive and hydrophobic polymers is connected to increased drug bioavailability.

The authors conclude that among the several systems present in the literature, tablets and films are the most common. In particular, unidirectional films can increase the efficacy of encapsulated peptides. 

The manuscript is concisely written, well documented and of interest to both the cognizant and non-cognizant reader.

AUTHOR RESPONSE: We thank the reviewer for the positive comment and appreciation of our manuscript.

Reviewer 3 Report

The manuscript prepared by Pratap-Singh et al. represents a review covering state of art in the field of buccal delivery of peptides for diabetes therapy. Overall, the manuscript is well written and structured. However, minor revision is required to avoid misleading the readers.

1. The type of article is misleading. It should be “Review” instead of “Communication”.

2. Section 2. What about C-peptide (or connecting peptide), which is known to be effective in the treatment of type II diabetes?

3. Section 3.2. The text in lines 359-410 is poorly divided into subsections. Combining different polymer properties, namely biodegradable hydrophobic PLA, PLGA, PCL and non-degradable hydrophilic pHPMA, in one subsection seems unreasonable. At the same time, the isolation of PLGA modified with a cell-penetrating peptide seems strange. The following organization of subsections in the indicated lines would be more rational:

“Biodegradable synthetic polymers and their modifications with PEG and cell-penetrating peptides” (text in lines 359-373 + 391-410).

“Non-biodegradable synthetic polymers” (text in lines 374-389).

4.      Line 541. It seems the reference number was failed. Please check this line.

Author Response

The manuscript prepared by Pratap-Singh et al. represents a review covering state of art in the field of buccal delivery of peptides for diabetes therapy. Overall, the manuscript is well written and structured. However, minor revision is required to avoid misleading the readers.

AUTHOR RESPONSE: We thank the reviewer for the appreciation of our work. We have carried out most of the recommendations proposed. We are so indebted for the reviewer's comments and invested time in the improvement of our manuscript.

  1. The type of article is misleading. It should be “Review” instead of “Communication”.

AUTHOR RESPONSE: We would like to address the concern raised regarding the classification of the article type. We understand the importance of accurately reflecting the nature of our work in the manuscript. While we do acknowledge that the content contains elements of a literature review, we would like to provide further context to clarify the rationale behind our choice of classification as a "Communication." The manuscript in question introduces the concept of a patent based on the reviewed literature. While the manuscript indeed contains a literature review, it serves as a foundation for the original work that was conducted subsequently, leading to the filing of a patent. The classification as a "Communication" aligns with our intention to present our original idea and intellectual property, while also providing an overview of the relevant literature that informed our work.

We recognize that the patent process involves considerations of novelty and obviousness. By classifying the manuscript as a "Communication," we aim to ensure that our original contribution is appropriately represented and that potential issues related to obviousness are addressed. Our primary objective is to avoid any conflicts with the patent application and to ensure the integrity of our intellectual property. We hope this clarification sheds light on our decision to classify the manuscript as a "Communication." We value your expertise and feedback, and we remain committed to accurately representing our work while adhering to the guidelines that best serve the interests of our research and intellectual property.

2. Section 2. What about C-peptide (or connecting peptide), which is known to be effective in the treatment of type II diabetes?

AUTHOR RESPONSE: Thank you for bringing up the valuable point about C-peptide's effectiveness in treating type II diabetes. We have included a mention of C-peptide within the context of the discussion of peptides for diabetes therapy, ensuring a comprehensive coverage of relevant therapies. We have added: "Apart from these, C-peptide, also known as connecting peptides, also hold special importance in diabetes therapeutics. Co-secreted with insulin from pancreatic β-cells in equimolar amounts, C-peptides reflect the precursor molecule's cleavage during insulin biosynthesis and have been used historically as a marker for insulin release. New emerging evidence suggests that C-peptide exerts significant bioactive effects on cellular processes critical for glucose homeostasis and vascular function. The consideration of C-peptide as a therapeutic agent holds particular significance in the context of type II diabetes, where insulin resistance and β-cell dysfunction often coexist."

3. Section 3.2. The text in lines 359-410 is poorly divided into subsections. Combining different polymer properties, namely biodegradable hydrophobic PLA, PLGA, PCL and non-degradable hydrophilic pHPMA, in one subsection seems unreasonable. At the same time, the isolation of PLGA modified with a cell-penetrating peptide seems strange. The following organization of subsections in the indicated lines would be more rational: “Biodegradable synthetic polymers and their modifications with PEG and cell-penetrating peptides” (text in lines 359-373 + 391-410). “Non-biodegradable synthetic polymers” (text in lines 374-389).

AUTHOR RESPONSE: I genuinely appreciate your suggestions for improving the organization of the content in Section 3.2. Your proposed subsection divisions seem logical and will enhance the clarity of the discussion on various polymer properties and modifications. We have modified the titles now:

Biodegradable synthetic polymers and their modifications with PEG and cell-penetrating peptides: Synthetic polymers such as poly(lactic-co-glycolic acid) (PLGA), poly(ε-caprolactone) (PCL), polyethyline glycol (PEG) and polylactide (PLA) have a more diverse structure and higher mechanical strength, than natural counterparts, making them excellent candidates for oral peptide drug delivery [93]. Researchers have prepared PLGA-PEG NPs for the oral delivery of insulin. Small Ins-NPs can be formed by dissolving insulin and PLGA–PEG molecules in an organic phase (DMSO) with a mean hydrodynamic diameter of 150 nm and an insulin load over 10% and around 90% of conjugation efficiency [94]. The NPs described in this study were decorated with an engineered human albumin variant that had improved human FcRn binding, resulting in a 2-fold increase in transcytosis across polarized epithelia compared with NPs decorated with wild-type albumin [94]. When tested for oral delivery in human FcRn transgenic mice with induced diabetes, these NPs could reduce glycemia by approximately 40% after 1 hour, improving pharmacologic availability [94].

 In addition, combining natural and synthetic polymers can enhance the oral bioavailability and therapeutic efficacy of peptide drugs in diabetes mellitus. For example, PLGA can achieve a controlled release of peptide drugs and protect them from enzymatic degradation in the GI tract. But negatively charged PLGA NPs face limited penetration through the mucus and epithelial cell layer. To overcome this limitation, Araújo et al. use microfluidics technology to encapsulate GLP-1-loaded PLGA NPs and a DPP4 inhibitor (iDPP4) with an enteric HPMC-AS (hydroxypropylmethylcellulose acetylsuccinate) polymer to increase intestinal permeation [96]. In STZ-nicotinamide induced T2DM rats, the dual-delivery H-PLGA particles reduced blood glucose levels by 44% and remained constant for another 4 h after oral administration [96]. This process results in the possibility of using natural and synthetic polymers in designing oral peptide drug delivery systems with enhanced efficacy in treating diabetes mellitus. The combination of PLGA NPs with CS and CPP improves intestinal permeation, increasing the potential for successful oral delivery of peptide drugs. These findings have implications for improving the therapeutic options for diabetes and other diseases requiring oral peptide drug delivery.

Non-biodegradable synthetic polymers like pHPMA: poly-N-(2-hydroxypropyl) methacrylamide (pHPMA) is a non-biodegradable hydrophilic synthetic polymer that is frequently utilized in oral peptide drug delivery for diabetes mellitus treatment. pHPMA can serve as a mucus-inert coating material, which facilitates mucus permeation. A study by Liu et al. provides evidence for the potential of using mucus-inert polymer-coated mucoadhesive NPs to improve oral drug delivery. The muco-bioadhesive strength surges with the molecular weight, mucoadhesive polymer concentration, chain flexibility, presence of hydrogen bond-forming groups (hydroxyl, carboxyl, amines and amides), positively or negatively charged groups and reduced crosslinking density [91]. The pHPMA coating allows for excellent mucus permeability and efficient interaction with the underlying epithelial cells, facilitating the paracellular transport of the loaded drug via the opening of tight junctions [95]. In diabetic rats, the developed NPs exhibited a remarkable hypoglycemic response and increased serum insulin concentration after oral administration [95]. They highlight the importance of considering mucosal tissue, including the secreted mucus layers, as complex absorption barriers, and results demonstrate the feasibility of overcoming these barriers in a multi-step process. Furthermore, the extensive use of HPMA polymer as a drug carrier makes this delivery platform a promising candidate for clinical translation.

4. Line 541. It seems the reference number was failed. Please check this line.

AUTHOR RESPONSE: Thank you for pointing out the reference number error. We have corrected this.